# The Yin and Yang-Like Clinical Implications of the *CDKN2A/ARF/CDKN2B* Gene Cluster in Acute Lymphoblastic Leukemia

**DOI:** 10.3390/genes12010079

**Published:** 2021-01-09

**Authors:** Celia González-Gil, Jordi Ribera, Josep Maria Ribera, Eulàlia Genescà

**Affiliations:** 1Josep Carreras Leukaemia Research Institute (IJC), Campus ICO-Hospital Germans Trias i Pujol, Universitat Autònoma de Barcelona (UAB), 08916 Badalona, Spain; cgonzalez@carrerasresearch.org (C.G.-G.); jribera@carrerasresearch.org (J.R.); jribera@iconcologia.net (J.M.R.); 2Clinical Hematology Department, ICO-Hospital Germans Trias i Pujol, 08916 Badalona, Spain

**Keywords:** acute lymphoblastic leukemia, del(9p21.3), prognosis, leukemogenesis, treatment

## Abstract

Acute lymphoblastic leukemia (ALL) is a malignant clonal expansion of lymphoid hematopoietic precursors that exhibit developmental arrest at varying stages of differentiation. Similar to what occurs in solid cancers, transformation of normal hematopoietic precursors is governed by a multistep oncogenic process that drives initiation, clonal expansion and metastasis. In this process, alterations in genes encoding proteins that govern processes such as cell proliferation, differentiation, and growth provide us with some of the clearest mechanistic insights into how and why cancer arises. In such a scenario, deletions in the 9p21.3 cluster involving *CDKN2A/ARF/CDKN2B* genes arise as one of the oncogenic hallmarks of ALL. Deletions in this region are the most frequent structural alteration in T-cell acute lymphoblastic leukemia (T-ALL) and account for roughly 30% of copy number alterations found in B-cell-precursor acute lymphoblastic leukemia (BCP-ALL). Here, we review the literature concerning the involvement of the *CDKN2A/B* genes as a prognosis marker of good or bad response in the two ALL subtypes (BCP-ALL and T-ALL). We compare frequencies observed in studies performed on several ALL cohorts (adult and child), which mainly consider genetic data produced by genomic techniques. We also summarize what we have learned from mouse models designed to evaluate the functional involvement of the gene cluster in ALL development and in relapse/resistance to treatment. Finally, we examine the range of possibilities for targeting the abnormal function of the protein-coding genes of this cluster and their potential to act as anti-leukemic agents in patients.

## 1. Introduction

Acute lymphoblastic leukemia (ALL) is a malignant clonal expansion of lymphoid hematopoietic precursors that exhibit developmental arrest at varying stages of differentiation, thereby partially recapitulating normal lymphoid ontogeny. Two subtypes are defined, according to which lymphoid progenitor is affected: B-cell-precursor ALL (BCP-ALL) and T-cell ALL (T-ALL). The incidence of ALL differs with age, whereby there is an early peak at 4 to 5 years (incidence of four to five per 100,000 people per year), a decline in young adults, followed by a slight increase after 50 years of age (incidence of up to two per 100,000 people per year) (www.seer.cancer.gov/statistics). Survival rates are lower in adults than in children. The improvement of treatment protocols over the last ten years has transformed pediatric ALL into a highly curable disease with long-term survival rates above 90% [1]. In contrast, long-term adult overall survival (OS) is 35% to 45% [2].

Similar to what occurs in solid cancers, transformation of normal hematopoietic precursors is governed by a multistep oncogenic process that drives initiation, clonal expansion and metastasis. In this process, alterations in genes encoding proteins that govern processes such as cell proliferation, differentiation, and growth provide us with some of the clearest mechanistic insights into how and why cancer arises. In such a scenario, deletions in the 9p21.3 cluster involving *CDKN2A/ARF/CDKN2B* (hereafter *CDKN2A/B*) genes arise as one of the oncogenic hallmarks of ALL. Deletions in this region are the most frequent structural alteration in T-ALL and account for roughly 30% of copy number alterations found in BCP-ALL. The proteins encoded by the *CDKN2A/B* genes belong to the INK4 family of CDK inhibitors, which block the ability of the tandem cyclin D-CDK4/CDK6 kinases to inactivate Retinoblastoma (RB) growth-suppressive functions [3]. The founding member is P16-INK4a [3]. Intriguingly, the CDKNA/B locus encodes a second, structurally and functionally unrelated protein, the alternative reading frame (ARF) or P14^ARF^, which is also a potent tumor suppressor [4,5]. The ARF protein activates TP53 by binding directly to the TP53-negative regulator, MDM2 [6,7]. Thus, one locus encodes two proteins that functionally interface with RB and TP53, which are two other key tumor suppressors that drive oncogenesis.

Here, we review the literature concerning the involvement of the 9p21.3 locus containing the *CDKN2A/B* genes as a prognostic marker of good or bad response in the two ALL subtypes (BCP-ALL and T-ALL). We compare frequencies observed in studies performed on several ALL cohorts (adult and child), which mainly consider genetic data produced by genomic techniques. We also summarize what we have learned from mouse models designed to evaluate the functional involvement of the gene cluster in ALL development and in relapse/resistance to treatment. Finally, we examine the range of possibilities for targeting the abnormal function of the protein-coding genes of this cluster and their potential to act as anti-leukemic agents in patients. It is important to note that the genetic studies in ALL have mostly analyzed the impact of the locus, rather than the specific contribution of the *ARF* gene. However, the functional studies highlighting the contribution of the locus in ALL leukemogenesis rely on the specific role of the ARF protein in this disease.

## 2. Genetic and Epigenetic View of the *CDKN2A/B* Gene Cluster in ALL

### 2.1. CDKN2A/B Gene Cluster Organization and Transcripts

The cyclin dependent kinase inhibitor 2A (*CDKN2A*) gene, also known as *INK4A* or *P16-INK4A*, and its paralog, cyclin dependent kinase inhibitor 2B (*CDKN2B*), or *INK4B* or *P15-INK4B*, are located on chromosome nine in the 9p21.3 cytogenetic band (information at https://www.ncbi.nlm.nih.gov/gene. ID: 1029). The two genes are arranged in tandem in the adjacent DNA and are transcribed on the anti-sense strand (Figure 1A). The protein products of these genes, P16 and P15, are almost identical in their structure and biochemical properties and act as specific inhibitors of CDK4/6 kinases [3,7], suggesting that the genes arose from a duplication event during evolution. *CDKN2A* generates several transcript variants that add a level of complexity and diversity to this gene cluster. Up to 14 different transcripts have been identified in silico, including protein-coding genes (isoforms) and non-coding RNA (information at https://www.ensembl.org/Homo_sapiens/Gene, ID: *CDKN2A* ENSG00000147889). However, only three alternatively spliced variants encoding proteins have been cloned from cells and demonstrated to be functional, two of which, P12 and P16γ, are structurally related isoforms that act as inhibitors of CDK4 kinase. High levels of *P16γ* expression are detected in primary T-ALL samples and in neuroblastoma cell lines [8]. The *P12* transcript is exclusively expressed in the human pancreas [9]. The third transcript, the alternative reading frame (*ARF*) gene, also known as *P14^ARF^*, is produced from the two alternative first exons joined to the *CDKN2A* exon two at the same acceptor site but in a different reading frame, resulting in a completely different protein [4] (Figure 1B).

In the case of *CDKN2B*, two distinct transcripts are generated by alternative splicing. One of these is a non-coding protein (information at https://www.ensembl.org/Homo_sapiens/Gene, ID: *CDKN2B*ENSG00000147883), while the other, *P10*, is an alternatively spliced transcript of *CDKN2B* that is ubiquitously expressed in normal and tumor cell lines [10]. The P10 protein product arises from a splicing defect in the 5′ donor site of intron one of *CDKN2B*, followed by a stop codon that is 79 nucleotides from the normally used splice site junction [10] (Figure 1B).

### 2.2. 9p21.3 Deletion in ALL

#### 2.2.1. Methods to Detect the Alteration and Possible Origin

The established methods for detecting structural alterations in hematology are karyotyping and fluorescence in situ hybridization (FISH). These allow detection of structural and numerical alterations with sizes of >5 Mb and >150 kb, respectively. However, both techniques are disadvantaged by their limited resolution, and often provide only a partial view of the full spectrum of alterations present in ALL patients. In the last 15 years, the use of high-throughput techniques such as the comparative genomic hybridization array (CGHa), the single nucleotide polymorphism array (SNPa) or, most recently, the next generation sequencing (NGS) technique, have helped refine frequencies of this alteration in BCP-ALL and T-ALL acute leukemias.

It has been postulated that illegitimate function of the recombination-activating gene (RAG) complex, whose normal physiological activity mediates V(D)J recombination, may be behind the deletions in the *CDKN2A/B* gene cluster and the many other recurrent ALL deletions, for example, *IKZF1 (IKAROS)*, a key transcription factor that regulates the commitment of hematopoietic precursors in B cells [11]. The breakpoints of these deletions often localize in the recombination signal sequence (RSS) that is recognized by the RAG. The structure of the junctions is compatible with typical RAG double-strand DNA breaks [12,13,14,15,16]. However, the RSS sequences have not been found in all the breakpoints studied, raising the possibility that, in a small subset of lymphoid leukemias, the 9p21.3 deletions are caused by a mechanism other than illegitimate V[D]J recombination [13,14].

#### 2.2.2. del(9p21.3) in BCP-ALL

In BCP-ALL as a whole, *CDKN2A/B* deletions are the most common secondary genetic event. A preferential loss of the maternal allele has been documented, suggesting that germline variants in that allele may be behind these large loss biases [17]. More importantly, del(9p21.3) is correlated with a lower level of gene expression, even in patients with hemizygous losses [18,19]. Overall, the frequency of *CDKN2A/B* losses ranges from 15% to 35% in children and from 30% to 45% in adults, the losses being more frequent in high-risk patients of all ages (Table 1 and Table 2). The relatively low frequency of the *CDKN2A/B* deletion within the ETV6-RUNX1 and the high hyperdiploidy subgroups, which are both more prevalent in pediatric than in adult BCP-ALL, together with the high frequency of the deletion seen in BCR-ABL1, which characterizes a frequent genetic subgroup found in adult cases, may account for these age-related differences [20]. However, other studies have found no differences in the incidence of *CDKN2A/B* loss between children and adults [21,22]. Other cytogenetic subgroups in which *CDKN2A/B* deletions are more prevalent in BCP-ALL are the Philadelphia chromosome (Ph)-positive (Ph^+^) [23,24,25,26,27], the Ph-like [25,27,28,29], the IGH-ID4 [30] and the PAX5 P80R [31,32] subtypes. More recently, a significant association has been identified between *CDKN2A/B* losses and *IKZF1* deletions in Ph^+^ patients [23] and in Ph-negative (Ph^−^) patients associated with *JAK2* mutations [19,27,33,34,35]. However, the most frequent concomitant alterations in patients with del(9p21.3) are *PAX5* deletions due to the recurrent losses of 9p [27].

BCP-ALL presents with roughly equal proportions of heterozygous and homozygous *CDKN2A/B* deletions. In addition, the presence of multiple clones harboring heterozygous and/or homozygous losses at diagnosis, and other clones with wt *CDKN2A/B*, has been noted [76,77]. This clonal heterogeneity masks the results obtained by techniques that use bulk leukemia, such as multiplex ligation-dependent probe amplification (MLPA) or SNPa, and raises the question of whether, for instance, homozygous losses may be more critical to BCP-ALL progression than monoallelic ones [78,79,80]. Finally, in approximately 80% of cases, the minimum deleted region seen in BCP-ALL patients affects both genes. In the other 20% of cases, there is selective loss of one of the two genes, or simultaneous loss of both *CDKN2A/B* genes but at different gene dosages (monoallelic vs. biallelic deletion) (Programa Español de Tratamiento en Hematología (PETHEMA) group; data not published).

#### 2.2.3. del(9p21.3) in T-ALL

*CDKN2A/B* gene deletions are also the commonest alteration in T-ALL. The first study that reported this finding, based on Southern blot analyses, showed *CDKN2A/B* deletions in 70% of T-ALL cases, where they occur as homozygous deletions [81]. Despite the technical limitation, it was evident that this alteration plays a basic role in T-ALL, for which reason it has been studied over the years in pediatric and adult cohorts using different techniques.

Focusing on pediatric cohorts and using different techniques (SNP arrays, multiplex ligation-dependent probe amplification (MLPA) and digital MLPA), the frequency of *CDKN2A/B* deletions has been found to range from 50% to 81% (Table 1). Some studies of adults, who account for 25% of ALL cases, have used fluorescent in situ hybridization (FISH) to identify *CDKN2A/B* deletions (Table 2). Globally, these studies have shown a frequency of *CDKN2A/B* deletions of around 42%. More recently, the use of SNPs to study CDKN2A/B deletions has yielded frequencies between 28% and 50% (Table 2). Age stratification in ALL patients gives rise to a third group of patients of intermediate age, between children and adults, known as AYAs (adolescents and young adults). This group often presents unique specific genetic alterations [21]. The frequency of *CDKN2A/B* deletions in this age-related group accounts for 47% [21,82].

With respect to homozygosity, most deletions in this gene are present in both alleles (approximately 70% of cases), independently of the cohort age (Table 1 and Table 2). This observation is at odds with a study hypothesizing that biallelic deletions are more frequent in adults than in pediatric cases, since conversion of monoallelic into biallelic deletion could require additional time [83].

Similar to what occurs in BCP-ALL, T-ALL also shows a specific association of *CDKN2A/B* deletions with a particular subgroup of patients, specifically with the non-immature T-ALL leukemias. Since the initial estimates of 27% and 77% of *CDKN2A/B* deletions in the early T-cell precursor ALL (ETP-ALL) and non-ETP patients (*p* = 0.0036) [84], respectively, several studies have produced results concordant with this association. Therefore, the *CDKN2A/B* deletion is a common alteration in cortical/mature T-ALL subtypes characterized by the overexpression of *TLX1* and *TLX3* [48,50,53,71,74,75], whereas the frequency in immature subtypes is significantly lower [48,84]. T-ALL subtypes characterized by the presence of *CDKN2A/B* deletions also show a high frequency of the *NOTCH1* mutation, although this association is not statistically significant [85,86]. However, there is a subgroup, of 1–6% of adult and childhood T-ALL that is characterized by the presence of an *MYC* translocation that is associated with high rates of *CDKN2A/B* deletions (75%). The genetic subgroup is also associated with *PTEN* inactivation and the absence of *NOTCH1* and *FBXW7* mutations [87,88].

### 2.3. Epigenetic Modifications at the CDKN2A/B Gene Promoter (T-ALL and BCP-ALL)

Alterations in the methylation pattern of the promoter of the CDKN2A/B genes have also been described in ALL, although they are much less frequent than deletions. A review of the literature regarding this topic indicates a greater degree of promoter hypermethylation of these genes in T-ALL than in BCP-ALL (Table 3 and Table 4).

Globally, if we consider the B and T subtypes in the adult and pediatric cohorts together we find that the range of methylation is between 10% and 47% for the *CDKN2B* gene promoter and between 0% and 41% for the *CDKN2A* promoter (Table 3 and Table 4). These differences do not vary with age (25% pediatric vs. 31% adult cases for the *CDKN2B* gene promoter; 12% pediatric vs. 3% adult cases for the *CDKN2A* gene promoter) [99]. In BCP-ALL, *CDKN2B* hypermethylation is more frequent than *CDKN2A* hypermethylation, and methylation of both genes may also increase with age (Table 4). In T-ALL patients, we observe that the percentage of promoter methylation in the *CDKN2B* and *CDKN2A* genes ranges between 46% and 68%, and between 0% and 12%, respectively, in pediatric cohorts (Table 3). Little information is available for adult T-ALL cohorts and shows that the percentage of *CDKN2B* gene promoter methylation varies from 16% to 49%, and is 1% for the *CDKN2A* promoter (Table 4). In T-ALL, the *CDKN2B* methylation status is associated with an immature immunophenotype [70] and with ETP-ALL features [75].

### 2.4. Germline Predisposition Variants in the CDKN2A/B Gene Cluster (T-ALL and BCP-ALL)

Germline mutations in both genes, but most importantly in *CDKN2A*, have been identified by SNPa. These inherited variants are associated with an increased risk of suffering ALL in pediatric case–control studies [100], raising the question about whether these variants may also occur in adults, or if they are more critical at earlier stages of development. Conversely, SNPs that protect against BCP-ALL development have also been reported [101]. As well as *CDKN2A* coding region (exon) germline mutations, SNPs predisposing to BCP-ALL have been observed in introns [100] and in non-coding regions, such as its promoter, that are important for regulating *CDKN2A/B* gene expression [101].

A critical aspect of these variants is their preferentially familial inheritance. Once inherited, germline pathogenic variants have a clear preferential expression compared with the non-pathological allele and, importantly, are not affected by the recurrent *CDKN2A/B* deletions [102,103], suggesting that the two alterations, one in each allele, are both needed to fully disrupt the normal cellular function of P16 and P15, as has been shown for RB1 and TP53 in other cancer models [104]. No association has so far been reported among any particular ALL genetic subtype and *CDKN2A/B* polymorphisms or other polymorphisms affecting genes essential to ALL development. This may reflect the fact that germline ALL-predisposing SNPs, including those involving *CDKN2A/B*, *IKZF1* and *PAX5*, sustain a pre-leukemic environment favoring the appearance of primary genetic lesions that lead to leukemia, instead of causing the appearance of a specific rearrangement/genetic primary abnormality, at least when referring to *CDKN2A/B*-related germline variants [105]. However, a *CDKN2A* SNP specifically related to Down syndrome ALL patients has recently been reported [106].

## 3. Clinical Impact of *CDKN2A/B* Alterations in ALL

Given the range of frequencies of the deletion in the different ALL subtypes (B-ALL and T-ALL), it is reasonable to expect to find some differences in the impact of these changes in the clinical environment, so the results obtained will not necessarily be concordant with those obtained from analyzing mixed ALL cohorts. Similarly, the prognostic significance of *CDKN2A/B* deletions should be addressed through a consideration of the influence of age on patient outcome and the method employed to analyze the frequency of the alteration. In addition, the size of the study cohort can be an impediment to arriving at a more accurate prognostic value, especially if we want to estimate it for *CDKN2A/B* deletions within a particular cytogenetic subtype, or the combination of *CDKN2A/B* losses with other molecular alterations. Moreover, modern ALL treatment protocols include minimal residual disease (MRD) measurement for stratifying patients during treatment [107,108]. Therefore, the prognostic impact of genetic markers should be also assessed in combination with MRD values.

### 3.1. Clinical Implications of Deletions in BCP-ALL

It seems that the treatment optimization for children applied in the more modern MRD-oriented protocols may overcome the supposed poor outcome related to *CDKN2A/B* deletions. However, some evidence suggests that homozygous *CDKN2A* deletions may be specifically more damaging, even though patients are treated according to these modern protocols [19,35], especially in children without high-risk features [45] and in patients with early relapses [42]. Conversely, other authors have identified poorer-prognosis patients with heterozygous deletions (Table 1) [35]. There is very little information about the prognosis of these deletions within the AYA group in large series focusing on BCP-ALL. However, younger age may counterbalance the absence of *CDKN2A/B*, since there is no strong evidence of a link between this genetic marker and poor prognosis in this group of patients [44].

For adults, the prognostic impact of *CDKN2A/B* genes deletions is more evident in Ph^+^ than in Ph^−^ BCP-ALL patients. The paper by the German ALLcooperative group argues strongly that *CDKN2A/B* deletions are a reliable prognostic marker of poor prognosis in Ph^+^ patients treated with chemotherapy plus imatinib and allogeneic stem cell transplantation (allo-SCT) (Table 2) [26]. Results from previous studies were also in line with this observation [18,61,62]. The prognostic value of *CDKN2A/B* deletions is less clear in the case of Ph^−^ BCP-ALL, probably because the genetic background of Ph^−^ is much more heterogeneous than that of Ph^+^. On one hand, the UK group on ALL study suggests that *CDKN2A/B* losses have no impact on outcome [58], while on the other hand, analysis of smaller series of Ph^−^ patients suggests that *CDKN2A/B* deletions could be a marker of poor outcome, especially concomitantly with *IKZF1* [63,68,80] or *RB1* deletions [44], as has been shown in pediatric cohorts [109]. Frequent codeletion of *CDKN2A/B* and *IKZF1* (in addition to *RB1* deletion and *JAK/STAT* pathway mutations) has also been found in Ph-like patients, a new genetic subgroup recently identified by gene expression profiling (GEP) [110] and initially including the Ph^−^ group, suggesting that the worse outcome of this codeletion in Ph^−^ patients could be due to the negative impact of these deletions on Ph-like patients. Consistent with this, we have recently shown that *CDKN2A/B* deletions could also be a marker of poor prognosis in Ph-like patients (Table 2) [68].

Finally, very few studies have pointed out the importance of *CDKN2A/B* losses as a worse prognosis marker in MRD-oriented trials. We have shown that CDKN2A/B losses might be a marker of poor outcome independently of MRD in adult Ph^−^ patients treated according to the PETHEMA protocols [80], as has also been shown in some pediatric studies [19,35].

### 3.2. Clinical Implications of Deletions in T-ALL

Most studies of pediatric cohorts show that *CDKN2A/B* deletions have no prognostic relevance in T-ALL (Table 1), with the exception of the NOPHO (Nordic Society of Paedriatic Haematology and Oncology) cohort study, in which deletions in the *CDKN2A/B* gene cluster were associated with lower OS. However, no effects on event-free survival (EFS) or relapse-free survival (RFS) were observed [111]. In contrast, many studies have shown that, in terms of OS, the presence of deletions in *CDKN2A/B* genes confers a better outcome, or a trend towards one, in adult T-ALL patients (Table 2). The good outcome observed in adult T-ALL patients is consistent with the fact that deletions in *CDKN2A/B* are more frequent in cortical/mature T-ALL subgroups, which are characterized by their better outcome when compared with more immature subtypes [70,71,74]. The exception to that result is the UKALL cohort study, in which the OS was identical in patients with and without deletions [69]. It is of note that the difference in outcome revealed by the various studies was not related to the gene dosage (homozygous vs. heterozygous deletions) (Table 2).

Finally, if we consider the MRD values when the analysis of the impact of *CDKN2A/B* deletions is assessed we note that only the ALL Spanish Cooperative Group (PETHEMA) has analyzed this relationship. We showed that patients with biallelic or monoallelic deletions of *CDKN2A* have stronger MRD responses (MRD levels ≤ 0.1% at the end of induction treatment) than those with normal copy number values. Despite these findings, when independent prognosis factors for OS were sought in multivariate analyses, MRD after induction therapy proved to be the only variable with independent predictive value [74].

### 3.3. Clinical Impact of Epigenetic Modifications (BCP-ALL and T-ALL)

Unlike deletion, the prognostic impact of *CDKN2A/B* promoter hypermethylation, and, to an even lesser extent, gene body hypermethylation and hydroxymethylation, has not been thoroughly analyzed in BCP-ALL because of the greater extent of promoter hypermethylation in T-ALL (Table 3 and Table 4). Accordingly, methylation of these genes does not seem to be very critical for BCP-ALL progression, and if so, this could be attributable to the combination of methylation and the loss of *CDKN2B* in the other allele [19,95]. However, it is surprising that only one study has explored the outcome of *CDKN2B* inactivation by methylation or deletion in T-ALL patients [75], showing that patients with either biallelic deletion or a high level of methylation exhibit lower 3-year EFS and OS than those with monoallelic deletion or low levels of methylation (Table 4).

Considering ALL globally (B-ALL and T-ALL subtypes), very few studies have analyzed the impact of methylation status in *CDKN2A/B* promoters. A study of childhood ALL showed that patients with a methylated *CDKN2B* promoter have a lower EFS rate and a higher incidence of relapse and mortality than those without methylation (Table 3) [93]. Conversely, in the only study of an adult ALL cohort, neither *CDKN2B* nor *CDKN2A* methylation affected the OS of patients (Table 4) [96].

## 4. Functional Implications of the *CDKN2A/B* Locus in ALL

INK4a, as a type of INK4 protein, binds to CDK4 and CDK6 and inhibits their kinase activity, thereby affecting RB function. The expression of *CDKN2A*, or of other family members, produces RB hypophosphorylation, which in turn leads to *E2F* repression and growth arrest. Absence of INK4a triggers constitutive RB phosphorylation and thereby E2F activation and growth progression [112]. However, ARF can also induce cell-cycle arrest, even in cells with active cyclin D, suggesting that RB-independent ARF signaling occurs that also controls cell-cycle arrest [4]. Studies done in *Arf*^+/+^ or *Arf*^+/−^ mouse embryonic fibroblasts (MEFs) showed that Arf and p53 form part of a common genetic pathway [113,114], revealing the relationship between these two tumors suppresses genes. Arf can inhibit the transformation of MEFs in the presence of MDM2 inhibitor (120) by directly binding to the MDM2 protein and inhibiting the ubiquitination of TP53, thereby stabilizing this tumor-suppressor protein [115,116,117,118]. Therefore, deletion in the *CDKN2A/B* locus simultaneously compromises the function of both RB and p53 tumor suppressors genes.

The first in vivo evidence that p16-INK4a (INK4a) and p14-ARF (ARF) can protect cells from acquiring oncogenic properties came from Ink4-null mice in which the expression of both genes (*Cdkn2a* and *Arf*) was eliminated [119]. These mice displayed, among others, features consistent with abnormal extramedullary hematopoiesis, suggesting that Ink4a and Arf normally regulate the proliferation of some hematopoietic progenitor cells [119]. However, this model was unable to resolve the oncogenic contribution of the individual proteins. The specific contribution of the ARF protein was assessed later in a single *Arf* KO. Mice lacking *Arf* expression were highly prone to spontaneous and carcinogen-induced tumors, including T cell lymphomas [113]. The mouse phenotype was much closer to that of double-null KO mice [119] than *Cdkn2a*-null mice [120,121], suggesting that the oncogenic properties associated with this locus were manly linked to the absence of the *ARF* gene.

### 4.1. Role of INK4a/ARF Proteins in Leukemogenesis

It has been suggested that the expression of *CDKN2A/B* genes varies during hematopoiesis [122], implying a possible role for these genes in leukemogenesis. The underlying idea is that *CDKN2A/B* genes would be epigenetically silenced by BMI1-containing polycomb repression complexes (PRCs) to facilitate both hematopoietic stem cell (HSC) and leukemic initiating cell (LICs) self-renewal. Absence of BMI1 would compromise the proliferative potential of leukemic stem and progenitor cells because they eventually undergo proliferation arrest and show signs of differentiation and apoptosis, leading to transplant failure of the leukemia. Defects resulting from BMI1 deletion can be partially rescued by co-deletion of *CDKN2A/B* genes, demonstrating the importance of maintaining silencing of this locus in early developmental stages of hematopoiesis and leukemogenesis [123,124,125,126,127].

With the aim to establish a functional relationship between constitutive *NOTCH1* signaling and *ARF* deletion in T-ALL, the hypothesis developed above was tested in NOTCH1-dependent T-ALL leukemias generated in mouse models [128]. Transformation of *Arf^+/+^* or *Arf*^−/−^ bone marrow precursor cells or thymocyte-derived cells with the constitutively active form of NOTCH1 (ICN1^+^) showed a bivalent H3k27me3 and H3k4me3 methylation pattern present throughout the locus in the *Ar*
^+/+^ and *Arf*
^−^^/−^ marrow-derived, and in the *Arf*^−/−^ thymocyte-derived cells. These modifications denote gene silencing [129] and detect binding of repressive Prc2 components (Ezh2 and Eed), which are known to participate in the repression of the *Cdkn2a/b* genes [123,124,125,126,127,128,129]. Promoters bearing bivalent H3K27Me3 and H3K4Me3 marks are thought to represent loci that are “poised” to begin transcription in response to appropriate stimuli. *Arf^-/-^* cultured thymocytes transduced with ICN1+ rapidly induced fatal T-ALL when infused into healthy syngeneic mice. In a similar way but with a long onset, ICN1+ bone marrow-derived progenitors ultimately gave rise to T-ALLs that were clinically and pathologically identical to those induced by thymocytes. In contrast, *Arf*^+/+^ ICN1^+^-transduced thymocytes expressed Arf protein and were less leukemogenic (135). This implies that in more mature T-cell progenitors the epigenetic remodeling of the *Arf* promoter is possible and, therefore, an additional genetic event in the *CDKN2A/B* gene locus, such as deletion, is needed to fully transform mature ICN1^+^ T-cell precursors [128,130].

The same hypothesis was tested in BCP-ALL. Expression of the BCR-ABL oncogene is the founding genetic lesion and the cytogenetic hallmark of both Ph^+^ ALL and chronic myeloid leukemia (CML) [131,132]. However, *CDKN2A/B* deletions do not occur in CML; probably because the leukemia arises from HSC-like progenitors [123,125], in which the *CDKN2A/B* locus is epigenetically silenced and “poised” to respond to an abnormally higher and sustained threshold of hyperproliferative signals [122]. Conversely, in Ph^+^ ALL, the leukemia-initiating cells appear to be committed lymphoid progenitors [133]. In that sense, mice engraftment of B-cell progenitors including the pro-B cells transduced with BCR-ABL1 oncogene showed that thus immature B-cell progenitors efficiently initiate Ph^+^ B-ALL, but pre-B cells did not do [134]. The reason of that is while in immature BCR-ABL1 transformed progenitors, Arf levels are maintained low or very low, in pre-B transformed cells are high and comparable to non-transformed controls [134,135,136]. Consistent with these results, the frequency of apoptotic cells in cultures initiated in transformed pre-B cells at 72 and 96 h after transduction were higher compared with pro-B cells [134]. Therefore, in order to bypass the BCR-ABL1-*Arf* expression re-activation loop, the more mature B-cell progenitors need to delete the *CDKN2A/B* locus to increase their oncogenic potential [23,137].

The matter of which upstream signals regulate *ARF* expression has also been explored in T-ALL, and given the similarly high level of co-occurrence with *NOTCH1* activating mutations, the possible relationship between ARF and NOTCH1 has been tested using null *Arf ^Gfp/Gfp^* thymocytes transduced with the ICN1^+^-CFP form. The study showed that a significant fraction of the CFP^+^ T-ALL cells co-expressed GFP, suggesting that the *ARF* gene can be activated by ICN1 signaling, albeit indirectly [130]. However, other results have shown that the *ARF* promoter can be activated before *NOTCH1* mutations are acquired [138]. The same study also evaluated the relationship between *ARF* expression activation and the LMO2 transcription factor (TF). Although the authors did not identify a direct role for LMO2 in inducing *ARF* expression, they did find that the TF could cooperate with *CDKN2A/B* loss to enhance self-renewal in primitive thymocytes [138]. In spite of this work, the activating stimuli that induce *ARF* expression under normal and leukemic conditions have not yet been elucidated.

### 4.2. Role of the INK4a/ARF Proteins in Genomic Instability

Unlike with hereditary cancers, sporadic cancers, such as ALL, show very few or no mutations in their DNA repair genes, suggesting that sporadic and hereditary cancers do not have the same etiology. Genomic instability could be induced by oncogenes instead of by the presence of mutations in DNA-repair genes. This hypothesis is based on the fact that analysis of NGS sequencing data have shown that very few genes are mutated, deleted and/or amplified at high frequencies in sporadic human cancers, those worth mentioning include the *TP53* tumor suppressor and DNA damage checkpoint gene and genes that negatively regulate cell growth, such as the *CDKN2A/B* genes. More importantly, very few or an absence of mutations in DNA-repair genes have been observed [139]. On the other hand, activation of growth signaling pathways induces loss of heterozygosity and genomic instability in mammalian cells cultured in vitro, human xenografts, mouse models [140,141,142,143,144]. These findings have led to the formulation of a mechanism by which activated oncogenes induced genomic instability involves DNA replication stress that preferentially affects common fragile sites [140,141,145]. In the context of leukemia, cells presenting *CDKN2A/B* deletions dysregulate cell-cycle, apoptosis and senescence-signaling pathways through TP53 and RB1. These tumor cells, with increased fast cycling, would accumulate additional mutations, thereby promoting clonal heterogeneity, drug resistance and tumor progression [76,137].

On the other hand, association of *CDKN2A* with telomere maintenance has been also observed. Maintenance of the in vitro growing of normal epithelial cells in a dish leads to a growth plateau in which most cells show proliferative arrest, while a small number of cells maintain good growth. These post-selected growing cells do not express *CDKN2A* mRNA and protein [146]. Continued proliferation of these cells leads to further telomere erosion, loss of the capping function, and entry into a phase of rampant chromosomal instability [147,148]. The massive genetic instability associated with this stage may well be the mechanism by which unusual cells acquire the constellation of genomic alterations needed for malignant transformation [147,149,150,151]. In a similar way, a correlation between *CDKN2A* expression and telomere length has also been described in patients with breast cancer, in whom repression of *CDKN2A/RB1* and/or *TP53/CDKN1A* by hypermethylation was associated with greater telomere shortening. Critical telomere shortening would lead to genome instability that ultimately produces malignant transformation [152]. Finally, more recent results have shown a TP53-independent role for *INK4a/ARF* at the mitotic checkpoint. Using MEFs without *Arf* expression, Britigan et al. have demonstrated that loss of *Arf* results in aneuploidy in vitro and in vivo. *Arf**^−/−^* MEFs exhibited mitotic defects including misaligned and lagging chromosomes, multipolar spindles, and increased tetraploidy. In addition, in these defective MEFs, overexpression of Mad2, BubR1, and Aurora B was observed. However, only overexpression of Aurora B phenocopied mitotic defects observed in *Arf*^−/−^ MEFs [153]. Despite these data, it is important to emphasize that the functional involvement of the *CDKN2A/B* gene cluster in telomere maintenance and mitotic check point regulation needs to be further explored in ALL.

### 4.3. Consequences of Germline Mutations

Very little is known about the functional consequences of germline mutations in the *CDKN2A/B* locus. However, it has been shown that these variants can modify protein-interacting domains in INK4a, affecting the interaction with other proteins like MYB [154], or leading to mislocalization of the INK4a protein into the cell nucleus [153].

## 5. Implications of the *CDKN2A/B* Gene Cluster for Treatment Resistance/Relapse

Comparison of the genetics in samples at diagnosis vs. relapse has helped to identify recurrent deregulated genes/pathways that are potentially responsible for relapse in ALL patients. In such an analysis, *CDKN2A/B* deletions are observed at diagnosis and at relapse, with a tendency to be more frequent homozygous deletions in ALL relapse cases [78,79,80,155,156,157,158]. However, some studies showed no significantly higher frequency of *CDKN2A/B* deletions (no homozygous or heterozygous deletions) at relapse than at diagnosis [24,38,159,160]. A higher level of *CDKN2A/B* promoter methylation during ALL progression has also been reported [96].

Another way of evaluating the oncogenic value of a specific genetic alteration is to look into the kinetics of the relapse of patients harboring that alteration. It is well known that patients experiencing early relapses respond less well to salvage therapy than those suffering late relapses. In BCP-ALL, *CDKN2A/B* deletions are significantly more closely related to early than to late relapses [161,162]. Some functional evidence corroborates these findings. It has been suggested that *CDKN2A/B* deletions could help attenuate treatment or facilitate resistance to tyrosine kinase inhibitors (TKIs) in mouse models. Arf inactivation could contribute to drug resistance by enhancing the maintenance of leukemia-initiating cells within the hematopoietic microenvironment (bone marrow), bestowing greater fitness on leukemic cells and facilitating the more rapid emergence of resistant leukemic clones expressing mutant BCR-ABL isoforms [163].

## 6. Therapeutic Approaches to Targeting the INK4 Tumor-Suppressor Protein Family

Due to the high prevalence of *CDKN2A/B* deletions in ALL patients and the fact that they are involved in regulating the cell cycle, we might have envisaged a potential use of INK4 family members as targets for exploring specific related therapies to treat ALL. However, this idea has been ruled out since these genes act as tumor suppressors in the cell. In spite of this, the regulatory function of INK4 proteins can be modulated via direct pharmacological inhibition of CDK4/CDK6 [164]. Consequently, selective and reversible inhibitors of CDK4/6 activity, such as palbociclib (PD0332991, Pfizer), ribociblib (LEE011, Novartis), and abemaciclib (LY2835219, Lilly), that block the cell cycle in the G1 phase and prevent leukemia progression are available and can be used to treat cancers with *CDKN2A/B* losses [165,166]. However, when the *RB1* gene is mutated, cyclin E1 and CDK2 become constitutively activated and leukemic cells become independent of the CDK4/6 pathway, which would render CDK4/6 inhibition ineffective [167]. Thus, selection of patients based on their RB mutational status is highly recommended in any clinical trial to gain efficacy from the use of CDK4/6 inhibitors.

Palbociclib (PD0332991, Pfizer) is an orally administered, small molecule inhibitor of CDK4/6 [168]. The molecule targets *Rb^wt^* tumor cells in vitro and in vivo, inducing G1 arrest by Rb phosphorylation and inhibition of E2f-dependent transcription [169]. Five clinical trials of palbociclib in ALL are currently underway (https://clinicaltrials.gov/). NCT03472573 is a phase I study testing the combination of palbociclib and dexamethasone in adults with recurrent and relapse BCP-ALL. Two trials (NCT03515200, NCT03792256) are testing the use of palbociclib in combination with various chemotherapeutic schedules in childhood ALL. Another clinical trial (NCT02310243) is assessing the dose and tolerability of the drug as a single agent in *MLL (KMT2A)* rearranged acute leukemias. The NCT03132454 trial is assessing the use of palbociclib alone or in combination with sorafenib, decitabine, or dexamethasone in recurrent and refractory acute leukemias.

LEE011 (Novartis) is an orally bioavailable small molecule that inhibits CDK4/6 at nanomolar concentrations [170]. Only one clinical trial is currently running, which is assessing the usefulness of the drug in combination with everolimus and dexamethasone in patients aged up to 30 years with refractory/relapse ALL (NCT03740334).

Abemaciclib, formerly known as LY2835219 (Eli Lilly), is the most potent orally available drug with the lowest enzymatic IC50, and like palbociclib and LEE011, is a small molecule that selectively targets CDK4/CDK6. Abemaciclib’s structure enables it to cross the blood–brain barrier at low doses and it may remain on-target for longer than palbociclib, as evidenced by orthotopic (intracranial) xenografts of glioblastoma cells [168]. No clinical studies are currently being conducted with this drug in ALL patients.

It is important to emphasize that the aforementioned clinical trials involving ALL focus on targeting BCP-ALL. This makes sense since, as we have explained in this review, deletions in the *CDKN2A/B* cluster give rise to distinct prognoses for the two ALL subtypes. Therefore, the selection of ALL patients tested in these clinical assays needs careful consideration.

## 7. Conclusions

Alterations in the *CDKN2A/B* gene locus arise as one of the hallmarks of ALL. The frequency of the deletion in this disease varies according to the specific ALL subtype, whereby it is more prevalent in T-ALL than in BCP-ALL, and to the age group, whereby it is more prevalent in pediatric T-ALL and adult BCP-ALL cases. Moreover, *CDKN2A/B* losses are associated with specific genetic lesions such as *IKAROS* deletions in BCP-ALL, or with the cortical subgroup in T-ALL. Surprisingly, these differences in frequency translate into a very different impact in the clinical environment. Specific association of this deletion with a particular subgroup with a marked prognosis impact (e.g., non-ETP-ALL and the Ph-like group) could be behind the contrasting clinical impacts of *CDKN2A/B* deletions in the BCP and T-ALL subtypes in general. In addition, the exact time during leukemogenesis when the alteration occurs may also influence the different clinical impacts of these deletions, in conjunction with some germline predisposition variants. However, the reasons why certain genetic associations present in certain patients in a particular time point of the leukemogenic process lead to different clinical outcomes are not well understood. To fill the gaps in our knowledge, we must delve deeper into the abnormal function that these genes jointly exert along the leukemogenic process. Therefore, the inclusion of more functional data to evaluate this will certainly deepen our understanding of the molecular bases of the yin and yang-like behavior of the *CDKN2A/B* deletions in ALL.

## Figures and Tables

**Figure 1 genes-12-00079-f001:**
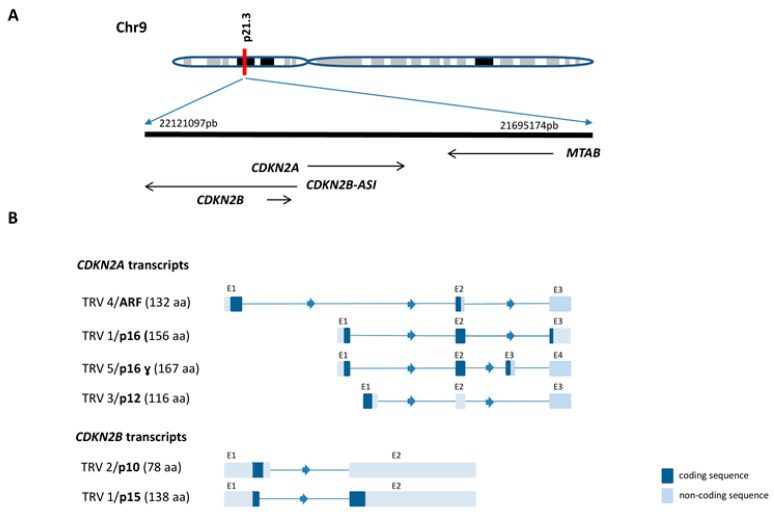
Genomic organization of the 9p21.3 locus and expressed genes (**A**) Localization and orientation of the *CDKN2A/ARF/CDKN2B* gene cluster (**B**) Alternative transcripts (and proteins) produced by different *CDKN2A* or *CDKN2B* promoter usage.

**Table 1 genes-12-00079-t001:** Frequency and clinical impact of the *CDKN2A/B* gene deletions in childhood acute lymphoblastic leukemia (ALL).

Reference	Trial or Patient Origin (Period)	Cohort Size	Age (y)	Type of ALL	Frequency del (Method)	EFS/DFS/RFS (p)	OS (p)	CIR (p)
[36]	CCG-1881, 1882, 1891, 1922 (1988 to 1995)	864	1–18	BCP	9p abn. 12% (Karyotyping)	Univariate: EFS 6y-9p abn 63% vs. no 9p abn 77% (*p* = 0.0004)	-	-
[37]	(1987–1997)	194	1–15	BCP	CDKN2A del homo 24%, ARF del homo 27%, CDKN2B del homo 18% (Southern blot, SSCP, Sanger sequencing)	Univariate: EFS CDKN2A del homo 0.58 vs. 0.77 (*p* < 0.001) Multivariate: CDKN2A del homo poor (*p* < 0.01)	-	-
[38]	DCOG ALL8 and 9 (1991–2004)	109	0–17	BCP	CDKN2A/B del 34% (FISH)	-	Univariate: 4y-CDKN2A/B del 80% vs. 87% (*p* = ns) Multivariate: CDKN2A/B del HR = 1.254 (*p* = 0.652)	Univariate: 4y-CDKN2A/B del 73% vs. 74% (*p* = ns) Multivariate: CDKN2A/B del HR = 1.251 (*p* = 0.608)
[39]	EORTC 58881 and 58951 (1989–2001)	227	0–17	BCP	CDKN2A del 31%, CDKN2B del 23% (qPCR)	Univariate: 6yEFS-CDKN2A del homo 68% vs. CDKN2A del hetero 80% vs. CDKN2A wt 75% (*p* = ns)	Univariate: CDKN2A del homo 82% vs. CDKN2A del hetero 90% vs. wt 87% (*p* = ns)	Univariate: CDKN2A del homo 11 vs. CDKN2A del hetero 5 vs. wt 32 (*p* = ns)
[40]	Disc. COG P9906 (2000–2003) Val. multiple COG protocols (1986–2007)	479	<18	Disc: 221 high-risk BCP Val: 258 BCP	CDKN2A/B del 46% disc. CDKN2A/B del 38% val. (SNPa)	Univariate: ns (outcome data not shown)	Univariate: ns (outcome data not shown)	Univariate: ns (outcome data not shown)
[41]	NOPHO2000 (2002–2006)	452	1–14	BCP	CDKN2A/B del 16% (FISH)	Univariate: 5yEFS-CDKN2A/B del homo 76% vs. CDKN2A/B del hetero 76% vs. CDKN2A/B wt 83% (*p* = 0.330)	-	-
[42]	ALL-REZ BFM 2002 (2002–2009)	294	0–18	BCP at 1st relapse	CDKN2A/B del 37% (MLPA)	Univariate: EFS CDKN2A/B del 0.45 vs. wt 0.43 (*p* = 0.990)	Univariate: CDKN2A/B del 0.48 vs. wt 0.54 (*p* = 0.443)	Univariate: CDKN2A/B del 0.40 vs. wt 0.21 (*p* = 0.001)
[43]	PETHEMA (1996–2014)	115	0–17	BCP	CDKN2A/B del 33% (CGH array)	Univariate: EFS ns (outcome data not shown)	Univariate: ns (outcome data not shown)	Univariate: ns (outcome data not shown)
[19]	ALL IC BFM 2002 and 2009 (2002–-2017)	641	2–12	BCP	CDKN2A del 26%, CDKN2B del 22% (MLPA, SNPa)	Univariate: RFS CDKN2A del homo HR 2.21 (*p* = 0.028) Multivariate: CDKN2A del homo HR = 3.09 (*p* = 0.007)	Univariate: 2y-CDKN2A/B del 85% vs. wt 88% (*p* = 0.560)	-
[44]	GIMEMA 2000-0904-1104-1308 and AIEOP ALL 2000, AIEOP-BFM ALL 2009 (2000–2018)	157	1–15 (*n* = 45)	BCP negative for BCR-ABL1, ETV6-RUNX1, TCF3-PBX1 or KMT2Ar	CDKN2A/B del 11% (MLPA)	Multivariate (children + AYA + adults): CDKN2A/B/RB1 HR = 2.12 (*p* = 0.048)	Univariate: ns (outcome data not shown)	Univariate: ns (outcome data not shown)
[45]	ANZCHOG ALL8 (2002–2011)	475	1–18	Non-high-risk BCP	CDKN2A/B del 36% (MLPA)	Univariate: 7y-EFS CDKN2A/B del homo 77% vs. del hetero 81% vs. wt 80% (*p* = ns)	Univariate: 7y-CDKN2A/B del homo 87% vs. del hetero 93% vs. wt 94% (*p* < 0.05)	Univariate: 7y-CDKN2A/B del homo 18% vs. del hetero 17% vs. wt 17% (*p* = ns)
[27]	DCOG-ALL10 (2004–2012)	515	1–18	BC	CDKN2A/B del 33% (MLPA)	Univariate: EFS CDKN2A/B del 79% vs. wt 87% (*p* = ns)	Univariate: ns (outcome data not shown)	Univariate: CDKN2A/B del 17% vs. wt 10% (*p* = ns)
[46]	ALLR3 (2003–2013)	192	1–18	1st (late) relapse BCP	CDKN2A/B del 22% (MLPA)	Univariate: 5y-CDKN2A/B del 63% vs. wt 62% (*p* = 0.75)	Univariate: 5y-CDKN2A/B del 69% vs. wt 75% (*p* = 0.26)	
[35]	ICICLE (Indian adaption of UKMRC2007 protocol) (2015–2017)	83	1–12	BCP	DKN2A/B del 36% (MLPA)	Univariate: 28month-EFS CDKN2A/B del 42% vs. wt 90% (*p* = 0.0004) Multivariate: CDKN2A/B del HR = 5.75 (*p* = 0.008)	-	-
[47]	St Jude Children’s Research Hospital (1993–2005)	50	<18	T-ALL	CDKN2A/B del 72% (SNP array)	-	-	-
[24]	UKALLXI ALL97-2003 (1986–2007)	266	<18	T-ALL	CDKN2A/B del 50% (SNP array, CGHa, FISH)	-	-	-
[48]	St Jude, the Children’s Oncology Group and AIEOP	ETP 42 Non-ETP 64	<18	T-ALL	ETP: CDKN2A del 25% Non-ETP: CDKN2A del 81% (SNPa)	-		Univariate. 5y-CDKN2A del 24.2% vs. wt 35.8% (*p* = 0.2814)
[49]	NOPHO ALL-1981–1986–1992–2000–2008 (1983–2011)	47	0-18	T-ALL	CDKN2A del 72% CDKN2B del 62.5% (SNPa)	Univariate: 5y-EFS CDKN2A del 0.48 vs. wt 0.73 (*p* = ns)	Univariate: 5y-CDKN2A del 0.52 vs. wt 0.91 (*p* = 0.04)	-
[50]	France and UK	155	111 c. 44 a.	T-ALL	CDKN2A del 78% (FISH, MLPA, CGHa, TDS)	-	-	-
[43]	PETHEMA (1996–2014)	27	<18	T-ALL	CDKN2A/B del 70.4% (CGHa)	Univariate: ns (outcome data not shown)	Univariate: ns (outcome data not shown)	Univariate: ns (outcome data not shown)
[51]	Children’s Oncology Group trial AALL0434 (2007–2011)	264	1–29	T-ALL	CDKN2A/B del 78.4% (SNPa)	Univariate: 5yEFS-CDKN2A del 90.6% vs. wt 92.7% (*p* = 0.349)	Univariate: 5y-CDKN2A del 94.5% vs. wt 100% (*p* = 0.0466)	Univariate: 5y-CDKN2A del 7.9% vs. wt 7.2% (*p* = 0.6953)
[52]	TPOG-ALL-93 (1995–2015)	102	<18	T-ALL	CDKN2A del 63.3%, CDKN2B del 50% (MLPA)	-	Univariate: ns (outcome data not shown)	Univariate: ns (outcome data not shown)
[53]	Brazilian Group Childhood Leukemia 99 (2005–2017)	341	<19	T-ALL	CDKN2A/B del 71.4% (MLPA)	-	Univariate: 5y-CDKN2A/B del 62.6% vs. wt 62.5% (*p* = 0.729)	-
[54]	Indian Childhood Collaborative Leukemia (ICICLE) (2017–2018)	27	<18	T-ALL	CDKN2A/B del 59.2% (digital MLPA)	-	Univariate: ns (outcome data not shown)	-

Y: years; EFS: event free survival; DFS: disease free survival; RFS: relapse free survival; p: probability; OS: overall survival; CIR: cumulative incidence of survival; BCP: B-cell precursor ALL; abn: abnormality; del homo: homozygous deletion; del hetero: heterozygous deletion; MLPA: multiplex ligation-dependent probe amplification; CGHa: comparative genomic hybridization array; SSPC: single-stranded conformation polymorphism analysis; ns: non-significant; HR: hazard ratio; disc: discovery cohort; val: validation cohort; c = children; a: adults; TDS: target deep sequencing; CCG: Children’s Cancer Group; DCOG: Dutch Childhood Oncology Group; EORTC: European Organization por Cancer Research; COG: Children’s Oncology Group; NOPHO: Nordic Society of Paedriatic Haematology and Oncology; FISH: Fluorescent In Situ Hybridization; ALL-REZ BFM: The German Berlin-Frankfurt-Münster study group on relapsed ALL; PETHEMA: Programa Español de Tratamiento en Hematología; ALL IC BFM: The German Berlin-Frankfurt-Münster intensive chemotherapy trial; GIMENA: Italian Group of Adult Hematological Diseases; AIEOP: Italian Association in Pediatric Hematology and Oncology; ANZCHOG: Australian and New Zealand Children’s Haematology/Oncology; TPOG: Taiwan Pediatric Oncology Group.

**Table 2 genes-12-00079-t002:** Frequency and clinical impact of the *CDKN2A/B* gene deletions in adult ALL.

Reference	Trial or Patient Origin (Period)	Cohort Size	Age (y)	Type of ALL	Frequency del (Method)	EFS/DFS/RFS (p)	OS (p)	CIR (p)
[55]	MRC UKALLXII/ECOG E2993 (1993–2004)	796	15–65	Ph^−^ BCP	del(9p) 9% (Karyotyping)	Univariate: 5y-EFS del(9p) 49%, O/E 0.73 (*p* = 0.043)	Univariate: 5y-del(9p) 58%, O/R 0.70 (*p* = 0.032)	-
[56]	L-10 and Swedish ALL group protocol (1986–2006)	240	17–78	BCP	9p abn. 7% (Karyotyping)	Univariate: median EFS 9p abn 6 months vs. no 9p abn 2.5 years, (*p* = 0.0134)	Univariate: median OS 9p abn 5 months vs. no 9p abn+ no HSCT 5y (*p* = 0.023) Multivariate: 9p abn RR = 2.21 (*p* = 0.032)	-
[57]	Japan Adult Leukemia Study Group (JALSG) (2002–2005)	80	15–64	Ph^+^ BCP	9p abn. 10% (Karyotyping)	Univariate: lower RFS, (*p* = 0.005)	-	-
[18]	GIMEMA LAL0201-2000 and LAL1205 (1996–2008)	101	18–76	Ph^+^ BCP	CDKN2A del 29%, CDKN2B del 25% (SNPa, FISH)	Univariate: 2y-DFS CDKN2A/B del 22% vs. wt 58% (*p* = 0.001) Multivariate: CDKN2A/B del poor DFS (*p* = 0.005)	Univariate: 2y-CDKN2A/B del 57% vs. wt 78% (*p* = 0.02)	Univariate: 2y-CDKN2A/B del 73% vs. wt 38% (*p* = 0.001)
[58]	UKALLXII/ECOG2993 (1993–2006)	454	15–65	Ph^−^ BCP	CDKN2A/B del 24% (MLPA, FISH)	Univariate: 5y-EFS CDKN2A/B del 39% HR = 1.20 (*p* = 0.247) 5y-EFS CDKN2A/B homo del vs. mono del HR = 0.59 (*p* = 0.08)	Univariate: 5y-CDKN2A/B del 42%, HR= 1.16 (*p* = 0.366)	-
[59]	PETHEMA AR93-03, OLD07, RI96-RI08 and Ph08 (1993–2013)	152	15–74	BCP	CDKN2A/B del 42% (MLPA)	-	Univariate: 5y-CDKN2A/B del 25% vs. wt 57% (*p* = 0.001); 5y-Ph^+^ CDKN2A/B del 14% vs. 54% (*p* = 0.025) Multivariate: CDKN2A/B del HR = 2.545 (*p* < 0.001)	Univariate: CDKN2A/B del 54% vs. wt 41% (*p* = 0.063); 5y-Ph^+^ CDKN2A/B del 100% vs. 43% (*p* = 0.071)
[60]	Chinese Han-South Medical University (2008–2013)	215	15–60	BCP	Diagnosis: CDKN2A/B del 28% 1st relapse: CDKN2A/B del 45% (FISH)	Univariate diagnosis: EFS CDKN2A/B del 12 vs. wt 24 months (*p* < 0.0001) Univariate 1st relapse: EFS CDKN2A/B del 5 vs. wt 16 months (*p* = 0.004)	Univariate diagnosis: CDKN2A/B del 19 vs. wt 30 months (*p* < 0.0001) Univariate 1st relapse: CDKN2A/B del 8 vs. wt 18 months (*p* = 0.001)	Univariate diagnosis: 2y-CDKN2A/B del 59% vs. wt 36% (*p* = 0.002)
[43]	PETHEMA AR93-03-11, RI96, OLD07, Ph00-08 (1996–2014)	100	18–84	BCP	CDKN2A/B del 47% (CGHa)	Univariate: ns (outcome data not shown)	Univariate: ns (outcome data not shown)	Univariate: ns (outcome data not shown)
[61]	Chinese Han-South Medical University (2008–2014)	135	18–65	Ph^+^ BCP	CDKN2A/B del 33% (FISH)	Univariate: 2y-DFS CDKN2A/B del 23% vs. wt 35% (*p* = 0.005)	Univariate: 2y-CDKN2A/B del 51% vs. wt 65% (*p* = 0.004)	Univariate: 2y-CDKN2A/B del 59% vs. wt 35% (*p* = 0.008)
[62]	Asan Medical Center, Korea. (2000–2015)	122	19–74	Ph^+^ BCP	del(9p) 20% (Karyotyping)	Univariate: 5y-DFS del(9p) 34% vs. wt 61% ( *p*= 0.189) Multivariate: DFS del(9p) HR = 3.42 (*p* = 0.002)	Univariate: 5y-del(9p) 44% vs. wt 76% (*p* = 0.091) Multivariate: del(9p) HR = 2.16 (*p* = 0.031)	-
[63]	Huntsman Cancer Institute (UT) and, Ann Arbor (MI) and Intermountain Healthcare (UT) (1998–2016)	70	18-83	BCP	CDKN2A/B del 49% (SNPa)	Univariate: median EFS CDKN2A/B del 9.5 months HR = 1.10 (*p* = ns)	Univariate: median OS CDKN2A/B del 21.8 months HR = 1.36 (*p* = ns); CDKN2A/B + IKZF1 del HR = 2.6 (*p* = 0.0007)	-
[64]	MD Anderson cohort (2001–2016)	182	19–85	Ph^+^ BCP	del(9p)-16% (Karyotyping)	Univariate: 5-y RFS del(9p) 34% (*p* = ns)	Univariate: 5y-del(9p) 26% (*p* = ns)	-
[65]	GRAALL 2003–2005 (2003–2011)	542	15–59	Ph^−^ BCP	del(9p) 12% (Karyotyping)	Univariate: EFS del(9p) HR = 1.05 (*p* = 0.78)	Univariate: del(9p) HR = 0.86 (*p* = 0.46)	Univariate: del(9p) SHR = 1.10 (*p* = 0.65)
[44]	GIMEMA 2000-0904-1104-1308 and AIEOP LLA 2000, AIEOP-BFM ALL 2009 (2000–2018)	157	15–35 (*n* = 56) 36–78 (*n* = 56)	BCP negative for BCR-ABL1, ETV6-RUNX1, TCF3-PBX1 or KMT2Ar	15–35 CDKN2A/B del: 48% 36–78 CDKN2A/B del: 46% (MLPA)	Univariate: 5y-DFS A. CDKN2A/B and/or RB1 del 13% vs. wt 54% (*p* = 0.03) Multivariate: (all ages): CDKN2A/B/RB1 del HR = 2.12 (*p* = 0.048)	Univariate: ns (outcome data not shown)	Univariate: ns (outcome data not shown)
[26]	GMALL 06/99 and 07/2003 (2001–2009)	97	18–64	Ph^+^ BCP	CDKN2A/B del 41% (SNPa, MLPA)	Multivariate: DFS CDKN2A/B del HR 2.621 (*p* = 0.0054)	Multivariate: CDKN2A/B del HR 2.162 (*p* = 0.014)	-
[66]	GIMEMA LAL0201B-0904-1205-1509 (2000–2018)	116	18–89	Ph^+^ BCP	CDKN2A/B del 32% (SNPa, MLPA)	3y-DFS IKZF1 + CDKN2A/B and/or PAX5 del 25% vs. 43% IKZF1 del only (*p* = 0.026) Multivariate: DFS CDKN2A/B del HR = 1.608 (*p* = 0.089)	3y-IKZF1 + CDKN2A/B and/or PAX5 del 40% vs. 63% IKZF1 del only (*p* = 0.02)	Univariate: ns (outcome data not shown)
[67]	PETHEMA AR93-03-11, OLD07, RI96-08 (1993–2017)	128	15–75	Ph^−^ BCP	CDKN2A/B del 44% (MLPA)	Univariate: 5-y DFS CDKN2A/B del 25% vs. wt 47% (*p* = 0.027)	Univariate: 5y-CDKN2A/B del 34% vs. wt 57% (*p* = 0.042) Multivariate: CDKN2A/B del HR = 2.216 (*p* = 0.023)	Univariate: 5-y CDKN2A/B del 56% vs. wt 41% (*p* = 0.090)
[68]	PETHEMA AR03 and AR11 (2003–2017)	44	16–59	BCP negative for BCR-ABL1, ETV6-RUNX1, TCF3-PBX1, KMT2Ar, high hyperdiploid and low hypodiploid	CDKN2A/B del 43% (MLPA)	Univariate: DFS CDKN2A/B del HR = 2.861 (*p* = 0.032) Multivariate: DFS CDKN2A/B del HR = 2.940 (*p* = 0.064)	Univariate: CDKN2A/B del HR = 2.523 (*p* = 0.073) Multivariate: CDKN2A/B del HR = 4.039 (*p* = 0.029)	Univariate: CIR CDKN2AB del HR = 2.900 (*p* = 0.039)
[69]	UKALL XII/ECOG 2993 (1993–2006)	108	>18	T-ALL	CDKN2A/B del 42% (FISH)	-	Univariate: 5y-CDKN2A del 52% (33;71)	-
[70]	GMALL 07/2003 and GMALL Elderly 01/2003	90	18–88	T-ALL	CDKN2A/B del 43% (FISH)	-	Univariate: 2y-CDKN2A/B del 77.2% vs. wt 47.2% (*p* = 0.076)	-
[71]	UKALL XII/ECOG 2993	53	>18	T-ALL	CDKN2A/B del 41% (CGHa)	-	Univariate: 5y-CDKN2A/B del homo 71% vs. del hetero 38% (*p* = 0.0119)	-
[72]	Lithuania (2007–2013)	25	18–64	T-ALL	CDKN2A/B del 28% (SNPa)	Univariate: ns (outcome data not shown)	Univariate: ns (outcome data not shown)	-
[43]	PETHEMA AR93-03, AR11, RI96, OLD07, Ph00-08 (1996–2014)	23	18–84	T-ALL	CDKN2A/B del 8.7% (CGHa)	Univariate: ns (outcome data not shown)	Univariate: ns (outcome data not shown)	Univariate: ns (outcome data not shown)
[73]	Institute of Hematology and Blood Diseases Hospital (China) (2009–2015)	18	14–61	T-ALL	CDKN2A del 50% CDKN2B del 33.3% (MLPA)	Univariate: ns (outcome data not shown)	Univariate: ns (outcome data not shown)	Univariate: ns (outcome data not shown)
[74]	PETHEMA HR-2003-11 (2003–2017)	62	16–72	T-ALL	CDKN2A del 50% CDKN2B del 47% (qPCR)	-	Univariate: 3y- CDKN2A/B del 75% vs. wt 36% (*p* = 0.05)	-
[75]	Seoul St. Mary’s Hospital (2004–2015)	102	2–77	T-ALL	CDKN2A/B del 45.1% (MLPA)	-	Univariate: ns (outcome data not shown)	-

Y: years; abn: abnormality; O/E: observed-to-expected; HSCT: Hematopoietic stem cell transplantation; RR: relative risk; del homo: homozygous deletion; del hetero: heterozygous deletion; ns: non-significant; HR: hazard ratio; EFS: event free survival; DFS: disease free survival; RFS: relapse free survival; p: probability; OS: overall survival; CIR: cumulative incidence of survival; BCP: B-cell precursor ALL; Ph^+^ BCP: Philadelphia chromosome positive BCP; Ph^−^ BCP: Philadelphia chromosome negative BCP; MLPA: multiplex ligation-dependent probe amplification; CGHa: comparative genomic hybridization array; ns: non-significant; HR: hazard ratio; disc: discovery cohort; val: validation cohort; c = children; a: adults; TDS: target deep sequencing; MRC: Medical Research Council; ECOG: Eastern Cooperative Oncology Group; UT: Utah; MI: Michigan; GRAALL: Group for Research on Adult Acute Lymphoblastic Leukemia trial; German Multicenter ALL: German Multicenter ALL trial.

**Table 3 genes-12-00079-t003:** Frequency and impact of the *CDKN2A/B* gene promoter methylation status in childhood ALL.

Reference	Type of ALL	Cohort Size	Age (y)	Technique	Frequency of Methylation		Prognosis	
					*CDKN2A* (*n*)	*CDKN2B* (*n*)	*CDKN2A*	*CDKN2B*
[89]	BCP	23	<18	MS-PCR	0% (23)	48% (23)	-	
	T-ALL	12			0% (12)	50% (12)	-	
[90]	T-ALL	45	<18	MS-PCR	11.7% (17)	68% (25)	-	
[91]	BCP	36	<18	MS-PCR	-	13% (23)	-	
	T-ALL					46.2 (13)		
[39]	BCP	227	0–17	MS-PCR	13% (31)	37.5% (28)	Non-significant	
[92]	BCP and T-ALL	95	<18	MS-PCR	4% (95)	25% (95) *	Non-significant	
[19]	BCP	333	<18	MS-MLPA	3.9% (333)	87% (333)	Non-significant	Univariate: trend to poor OS
[93]	BCP	93	1–13	MS-PCR	-	57% (21)	-	Univariate: EFS-8y hyper.71% vs. hypo 91% (*p* = 0.02); rate of relapse hyper 28% vs. hypo 9.3% (*p* = 0.02)
	T-ALL				-	38% (72)		

* Methylation of p15 gene occurred more frequently in T-ALL than in precursor B-ALL (*p* = 0.02) y: years; n: number of cases analyzed; BCP: B-cell precursor ALL; EFS: event free survival; OS: overall survival; MS-PCR: methylation specific PCR; MS-MPL: methylation specific MLPA; hyper: hyper methylation pattern; hypo: hypo methylation pattern.

**Table 4 genes-12-00079-t004:** Frequency and impact of the *CDKN2A/B* gene promoter methylation status in adult ALL.

Reference	Type of ALL	Cohort Size	Age (y)	Technique	Frequency of Methylation		Prognosis	
					*CDKN2A* (*n*)	*CDKN2B* (*n*)	*CDKN2A*	*CDKN2B*
[94]	BCP	41	>18	MS-PCR	12.5% (41)	2.4% (41)	-	Univariate: 5y-OS methy 12% vs. un-methy 36% (*p* = 0.84); 5y-DFS methy 7% vs. un-methy 19% (*p* = 0.98)
	T-ALL	8			62.5% (8)	39% (8)		
[95]	BCP	70	>18	MS-PCR	23% (70)	37% (70)	-	Multivariate: normal *CDKN2B* was a favourable prognostic factor for longer DFS (*p* = 0.0001)
[96]	BCP	80	>18	MS-PCR	2.5% (80)	22.5% (71)	Univariate: Ph^−^ (*n* = 57), 5y-OS methy 50% vs. un-methy 42% (*p* = 0.8)	Univariate: Ph^−^ (*n* = 57) 5y-OS methy 26% vs. un-methy 46% (*p* = 0.09)
	T-ALL						Non-significant	Non-significant
[97]	BCP and T-ALL	64	16–78	MS-PCR	-	25% (64)	-	Non-significant
[98]	Ph^−^ and MLL-BCP	199	15–83	Real Time bisulfite PCR	-	17.4% (189)	-	Non-significant
[70]	T-ALL	90	>18	MS-PCR	-	48.6% (74) *	-	-
[75]	T-ALL	102	2–77	pyrosequencing	3.8% (93)	50.6% (93) **	-	Univariate: 3y-EFS high methy 35.9% vs. low methy 59.1% (*p* = 0.042) Multivariate: *CDKN2B* biallelic deletion or high methylation HR = 6.358 (*p* = 0.012)

* *CDKN2B* methylation status was associated with the early immunophenotype subtype (*p* = 0.021). ** Most ETP-ALL cases were included in the *CDKN2B* hypermethylation group y: years; n: number cases analyzed; BCP: B-cell precursor ALL; EFS: event free survival; OS: overall survival; DFS: disease free survival; HR: Hazard ratio; MS-PCR: methylation specific PCR; methy: promoter methylation; un-methy: un-methylated promoter.

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
