# Peer review of "The Yin and Yang-Like Clinical Implications of the *CDKN2A/ARF/CDKN2B* Gene Cluster in Acute Lymphoblastic Leukemia"

_genes, 2021, doi:10.3390/genes12010079_

Round 1
Reviewer 1 Report
This is a comprehensive review on CDKN2A/B mutations in ALL on human patient prognosis and the impact of these mutations model systems.
Author Response
We thank you to the review 1 his/her positive evaluation.
Reviewer 2 Report
The manuscript "The Yin and Yang-like clinical Implications of the CDKN2A/ARF/CDKN2B Gene Cluster in Acute Lymphoblastic Leukemia", written by Gonzalez-Gil C, Ribera J, Ribera JM and Genesca E, describes the roles of CDKN2A/ARF/CDKN2B gene deletions in the development of different subtypes of acute lymphoblastic leukemia, as deletions on this locus present one of the oncogenic hallmarks of ALL. The manuscript is the review of the studies analysing the frequency and the clinical impact of CDKN2A/B gene deletions and downregulation in childhood and adult types of ALL.
The manuscript is well written, with a great amount of facts and two complex tables with all of data analysed. The data are well organized and systematized. The main topics cover organization and gene transcripts of the CDKN2A/B gene cluster, methods used for analysis, frequency of deletions and inactivation in different ALL types, clinical implications of the changes in the ALL, functions of CDKN in ALL, implications for prognosis and therapy. The paragraph describing functional implications of CDKN in ALL starts with the deletion consequences in mice and the role of ARF and INK in cell biology. Possibly these data could be put at the beginning of the article and be described more accurately (from 243 to 295). The lines 265-266 could be misunderstood. ARF is known as a tumour suppressor which can increase its expression as a consequence of oncogene activation. Oncogene activation in the normal cellular milieu increases activation of pRB and E2F in turn activates the ARF promoter. So, negative feedback circuit protects the cell by p53 activation and causes cycle arrest. ARF-deleted cells have advantage and proliferate (selection of clones). Also, the statement that genomic instability could be induced by oncogenes needs more explanation, especially because mechanisms causing breaks in replication stress are different from mechanisms involving recombination. Furthermore, p16 is involved in the senescence process in some cell types and pRb during these processes has a role in heterochromatinization of the telomeres. These processes (lines 284-288) should be more accurate described, to distinguish what is a direct act of p16, and what is a consequence of proliferation without telomerase activation. The sentence "these cells showed upregulation of Aurora Kinase B (AURKB) gene expression in other genes..." is not clear. So, biological mechanisms which involve p16 and ARF activities could be described in more details.
Minor:
line 82: affects
lines 92 and 93: ETV6-RUNX1 subtypes and BCR-ABL1 subtypes
line 160: "What little information... " change the sentence construction
line 258: delete quotation mark
line 298 "deletions prevail from diagnosis to relapse" change the sentence construction
line 342: ...deletions in BCP and T-ALL subtypes in general.
Tables 1-4: references should be put in the last colon and cited as a number
below should be written all the explanations of the shortcuts: from T-ALL, MLMA, CGH, p....
References should be written in the same way with all the data. For some references it is not necessary their availability to be written.
Author Response
“The manuscript is well written, with a great amount of facts and two complex tables with all of data analyzed. The data are well organized and systematized. The main topics cover organization and gene transcripts of the CDKN2A/B gene cluster, methods used for analysis, frequency of deletions and inactivation in different ALL types, clinical implications of the changes in the ALL, functions of CDKN in ALL, implications for prognosis and therapy”
Answer to the general comment: We thank the reviewer for the overall positive criticisms and we have nothing to add in this regard, except that the specific comments raised are addressed below in detail.
“The paragraph describing functional implications of CDKN in ALL starts with the deletion consequences in mice and the role of ARF and INK in cell biology. Possibly these data could be put at the beginning of the article and be described more accurately (from 243 to 295)”
Answer to the specific comment: Lines 243 to 295 corresponds to hole block of functional implications of CDKN2A/B in leukemia. Following the indications of the review we have extended this block to make it clearer and more understandable. Now this section expands from lines 241-325. However, in order to not alter the hole structure of the review, we have maintained this section after discussion of the clinical implications of the CDKN2A/B locus and before start the treatment part, as we consider that this is the natural/easy to understand order to explain the data.
“The lines 265-266 could be misunderstood. ARF is known as a tumor suppressor which can increase its expression as a consequence of oncogene activation. Oncogene activation in the normal cellular milieu increases activation of pRB and E2F in turn activates the ARF promoter. So, negative feedback circuit protects the cell by p53 activation and causes cycle arrest. ARF-deleted cells have advantage and proliferate (selection of clones)”
Answer to the specific comment: we have rewritten this part of the review to clearly explain how a tumor suppressor gene as it is ARF, can induce tumorigenesis and bypass its expression activation induced by the oncogene, in our case BCR-ABL1. The rewritten section corresponds to line 277 to 287.
“Also, the statement that genomic instability could be induced by oncogenes needs more explanation, especially because mechanisms causing breaks in replication stress are different from mechanisms involving recombination”
Answer to the specific comment: we thank you the review this appreciation. We have corrected it. In addition, we have tried to clarify how oncogenes could induce genome instability. The new text goes from lines 297-307.
“Furthermore, p16 is involved in the senescence process in some cell types and pRb during these processes has a role in heterochromatinization of the telomeres. These processes (lines 284-288) should be more accurate described, to distinguish what is a direct act of p16, and what is a consequence of proliferation without telomerase activation”
Answer to the specific comment: we thank you the review his comment, however in that section we have writing the facts, showed as simple associations, that the articles we comment on the review show. However, we have tried to explain this section clearer. The new version corresponds to line 308 to 316.
“The sentence "these cells showed upregulation of Aurora Kinase B (AURKB) gene expression in other genes..." is not clear. So, biological mechanisms which involve p16 and ARF activities could be described in more details”
Answer to the specific comment: following the instructions of the review we have explain this section more clearly. The new version corresponds to lines 316 to 321.
Minor:
line 82: affects
Answer to the specific comment: we have corrected the grammatical error. Highlighted in yellow.
lines 92 and 93: ETV6-RUNX1 subtypes and BCR-ABL1 subtypes.
Answer to the specific comment: We have changed the sentence to make it clearer. Highlighted in yellow.
line 160: "What little information... " change the sentence construction.
Answer to the specific comment: We have changed the sentence. Highlighted in yellow.
line 258: delete quotation mark.
Answer to the specific comment: We have done it. Highlighted in yellow.
line 298 "deletions prevail from diagnosis to relapse", change the sentence construction.
Answer to the specific comment: We have changed the sentence. Highlighted in yellow.
line 342: ...deletions in BCP and T-ALL subtypes in general.
Answer to the specific comment: We have changed the sentence. Highlighted in yellow.
Tables 1-4: references should be put in the last colon and cited as a number.
Answer to the specific comment: We have changed the way to cite the references in the tables, however in order do not altered the table format we have kept them at the beginning of the table.
below should be written all the explanations of the shortcuts: from T-ALL, MLMA, CGH, p....
Answer to the specific comment: We have checked all the abbreviations and included those that were missed below the table.
References should be written in the same way with all the data. For some references it is not necessary their availability to be written.
Answer to the specific comment: we have uniformized all the references along the text and remove those that were unnecessary.
Reviewer 3 Report
Minor
- All the text seems to be in this landscape format, I guess because of the configuration of the tables. This format should only be in the pages that have tables.
- Some acronyms appear undefined, such as BCP-ALL in the abstract or RB in the introduction.
- Figure 1: The figure legend should follow the figure and not before it.
- Tables 1 & 2: Because the tables are so large, it helps to have the first row, i.e. the name of the communities on all pages
Author Response
All the text seems to be in this landscape format, I guess because of the configuration of the tables. This format should only be in the pages that have tables.
Answer to the specific comment: thank you to the referee for this observation. We think that the personnel in charge of the edition will arrange this at their convenience.
Some acronyms appear undefined, such as BCP-ALL in the abstract or RB in the introduction.
Answer to the specific comment: we have revised the manuscript and add the definition of the acronyms that were missed.
Figure 1: The figure legend should follow the figure and not before it.
Answer to the specific comment: we have change that.
Tables 1 & 2: Because the tables are so large, it helps to have the first row, i.e. the name of the communities on all pages
Answer to the specific comment: thank you to the referee for the suggestion. We have tried it but we finally we not included in the new version because the extension of the table was compromised.